# Face Detection Using a Capsule Network for Driver Monitoring Application

**János Hollósi, Áron Ballagi, Gábor Kovács** ⓘ**, Szabolcs Fischer \*** ⓘ **and Viktor Nagy \***

Central Campus Győr, Széchenyi István University, 9026 Győr, Hungary; hollosi.janos@sze.hu (J.H.);
ballagi.aron@sze.hu (Á.B.); gkovacs@sze.hu (G.K.)
**\*** Correspondence: fischersz@sze.hu (S.F.); nviktor@sze.hu (V.N.); Tel.: +36-(96)-613-544 (S.F.)

**Abstract:** Bus driver distraction and cognitive load lead to higher accident risk. Driver distraction sources and complex physical and psychological effects must be recognized and analyzed in real-world driving conditions to reduce risk and enhance overall road safety. The implementation of a camera-based system utilizing computer vision for face recognition emerges as a highly viable and effective driver monitoring approach applicable in public transport. Reliable, accurate, and unnoticeable software solutions need to be developed to reach the appropriate robustness of the system. The reliability of data recording depends mainly on external factors, such as vibration, camera lens contamination, lighting conditions, and other optical performance degradations. The current study introduces Capsule Networks (CapsNets) for image processing and face detection tasks. The authors' goal is to create a fast and accurate system compared to state-of-the-art Neural Network (NN) algorithms. Based on the seven tests completed, the authors' solution outperformed the other networks in terms of performance degradation in six out of seven cases. The results show that the applied capsule-based solution performs well, and the degradation in efficiency is noticeably smaller than for the presented convolutional neural networks when adversarial attack methods are used. From an application standpoint, ensuring the security and effectiveness of an image-based driver monitoring system relies heavily on the mitigation of disruptive occurrences, commonly referred to as "image distractions," which represent attacks on the neural network.

**Keywords:** bus driver monitoring system; road safety; artificial intelligence; capsule network; face recognition

## 1. Introduction

The need for safe public transport is becoming more demanding for drivers as traffic density increases, accompanied by a complex traffic mix and an influx of fast external stimuli, particularly prevalent in urban areas. Advanced driver-assistance systems (ADASs) can partly intervene in emergencies, e.g., braking [1], but the driver's task and responsibility remains to control the vehicle safely. Even if ADASs' goal is to maximize safety, it should be considered that these complex systems have side effects. The adoption and utilization are difficult; a learning process and trust are needed [2]. The analysis of selected papers reveals that ADASs are associated with increased secondary task engagement and improved performance. In contrast, some papers indicate a diversion of attention from driving tasks, emphasizing the critical role of human drivers alongside vehicle automation and the significance of user understanding for the appropriate and effective use of ADASs [3]. Analysis of driver attention and cognitive load provides an opportunity to optimize working conditions and improve accident prevention solutions.

The penetration of public road transport in Europe is still significant. Ensuring passenger safety is a priority as vehicles carry more and more passengers in increasing traffic density. Although buses (in addition to trams and trolleys) account for a decreasing share of all modes of transport, their share is still 7.4% in the EU, based on 2020 data [4]. The

number of bus passenger-kilometers in the EU is more than 138 thousand million kilometers, after a 40% decrease due to COVID-19 (see Table 1) [5]. A review of historical data on accidents shows that the average number of fatalities in bus-related accidents per million inhabitants in the European Union countries is 1.3, based on 2017–2019 data. More analysts proved that with the increase in the number of motor vehicles, the number of traffic accidents with casualties also increase [6]. The victims of fatal accidents are mostly pedestrians, motorists, and bus passengers. Accident injury statistics (2010–2019) show that 2% of bus accidents involve serious injuries [4]. In the current study, Capsule Networks (CapsNets) were introduced as part of a professional camera-based Driver Monitoring System (DMS) for public transport. The study presented in this paper shows the initial results of a long-term research and development project.

**Table 1.** Motor coaches, buses, and trolley buses: European Union—27 countries [5].

| Year | 2013 | 2014 | 2015 | 2016 | 2017 | 2018 | 2019 | 2020 | 2021 |
|---|---|---|---|---|---|---|---|---|---|
| Percentage (%) | 10.4 | 10.0 | 10.0 | 10.0 | 9.6 | 9.5 | 9.5 | 7.4 | N/D |
| Passenger-kilometers (million km) | 238,368 | 230,616 | 246,661 | 240,574 | 224,813 | 225,958 | 231,099 | 135,427 | 138,262 |

Analysis of the data has identified many factors that influence the occurrence of accidents. Negligence of bus operators, driver failure, and external factors (weather and road conditions) were identified as the main causes [5]. Driver error can be broken down into several factors, typically depending on current mental and physical conditions, so monitoring driver behavior and later passive or active interventions can ensure safer bus transport [7]. Distractions are visual, auditory, physical, or cognitive stimuli that interfere with driving activities, as well as secondary activities, such as mobile phone use, communication with passengers, and daydreaming. If distraction can be defined as inattention, drowsiness can be defined as inattention related to the driver's physiological response. This biological behavior is typically associated with insufficient sleep, poor health, or long periods of driving in monotonous environments [8].

Thus, boredom, fatigue, monotony, and sleep deprivation increase the risk of accidents, as reduced attention can lead to impaired information processing and the inability to make decisions, leaving the driver unable to react in an emergency [9]. Distracted driving is "*any activity that diverts attention from driving, including talking or texting on your phone, eating and drinking, talking to people in your vehicle, fiddling with the stereo, entertainment, or navigation system—anything that takes your attention away from the task of safe driving*" [10]. Most research on vigilance focuses on sleep deprivation [11–13]. However, accident data and simulated driving studies suggest that a loss of alertness can also occur during the day, especially on long, monotonous roads [14]. Driver errors are more common during monotonous driving, with low task demand and stimulus levels, reducing attention. Furthermore, the personality profile of drivers also influences the likelihood of crashes due to hypervigilance; extroverts and sensation seekers are at higher risk. Various monitoring systems are developed and used to observe or detect driver behavior. Different approaches use psychological tests and different physiological sensors, data collection, and analysis techniques, as follows: (1) vehicle-based measures, (2) behavioral measures, and (3) physiological measures [15].

Vehicle-based measures were utilized in more research studies in order to identify driver behavior. One method was developed for detecting driving patterns using vehicle sensor data from a single turn [16]. CAN-BUS data analyzed by a unique deep learning framework resulted in driving behavior identification [17]. A human–vehicle interaction monitoring system with principal component analysis is used to monitor fuel consumption, $CO_2$ emissions, driving style, and driver's health in real-time with high prediction levels [18]. High energy efficiency targets are also important in the case of rail vehicles, where recent research focuses on detecting and optimizing energy losses on the vehicle side [19].

Several behavioral measurement analyses—mainly using the Driver Behavior Questionnaire—have shown that professional drivers engage in less risky behavior than non-professional drivers but are more likely to be involved in traffic accidents due to extended driving periods [20]. Some studies proved that driver characteristics are related to driving circumstances (e.g., comfort level) and affect driving performance [21]. A great sample of inter-city bus drivers was analyzed by psychometric properties of the Multidimensional Driving Style Inventory (MDSI), Driver Behavior Questionnaire (DBQ), and Driver Anger Scale (DAS) questionnaires and highlighted the importance of diagnostics for distinguishing safe and unsafe drivers [22]. Results show that the correlation between the data collected from the vehicle and drivers' signals (physiological sensors) is high, so it is expected that further studies based on the data collected from the vehicle will allow for high-accuracy inferences about driver behavior and driver identification [23].

Recently, it has been shown that the frequency and duration of fixation, the change in pupil diameter (pupillometry), the quantity of saccades and microsaccades, and the frequency and duration of blinking are closely related to cognitive load [24]. These measurements can be made with eye-tracking systems with high-quality cameras observing test participants' eye movements. Some studies using driving scene videos have shown that fixation behavior and patterns correlate with driving experience [25,26]. In other studies, average pupil size gave a clear indication of divided attention while driving after performing tasks of varying difficulty or with different interfaces on the IVIS (e.g., physical button or touchscreen), and it was observed that average pupil size increased significantly [27–29].

Driver fatigue and distraction detection can be based on biometric signals, steering motion, or driver's face monitoring [30]. Some researchers found that a higher average Heart Rate (HR) shows that the driver performs more demanding or secondary tasks [31]. Others performed heart rate variability analyses validated by electroencephalography for driver drowsiness detection [32]. More authors used wearable Galvanic Skin Responses (GSR) wristbands to identify the distraction of drivers during a driving experiment on-the-road with high accuracy and consistency [33]. Electroencephalograms (EEGs) are widely used for driver behavior detection, but their signals are not stationary, and the process is intrusive for testing participants, mainly when using multi-sensor systems [34]. Recently, single-channel EEG-based drowsiness detection systems were tested, where data analysis through small-sized time window datasets and a single feature computation enabled lower processing and storage capacities, leading to more straightforward implementation in embedded systems [35]. Other physiological sensors—like an electromyogram (EMG), which measures the driver's muscle fatigue—were also used, and the appropriate measurement methodology was described [36]. Emotion classification with high prediction accuracies has also been investigated, where an electrocardiogram (ECG)—combined with GSR—was applied [37].

The most widespread solution for DMSs is the non-wearable, camera-based system using computer vision for face recognition. This approach is more convenient for drivers because it does not require additional wearable devices, allowing for a seamless and non-intrusive monitoring experience. It can be perfectly integrated into the mass production process and thus added to series production vehicles. "*Attention warning in case of driver drowsiness or distraction*" must be applied to all new European vehicles from 2024, according to EU Regulation 2019/2144 [38]. Visual sensors, RGB, IR, or other cameras facing the driver, with different viewing angles, are used to collect naturalistic driving (ND) data, including drivers' interaction, behavior, and the surrounding (driver cabin). The following data categories are usually detected: hand movement, body pose estimation, face detection, and distraction/drowsiness [39]. Face detection methods are generally categorized as feature-based and learning-based methods [40]. Learning-based methods are usually more robust than feature-based methods but mostly take more computational resources. However, these methods can reach a more than 80% detection rate in laboratory conditions [41]. It is a challenge to test the applicability of the appropriate algorithm and to prove its benefits, but in the long term, optimized face recognition allows the development of a

system with a higher confidence level and increases road safety. Data recording and processing reliability depend on the following factors: viewing angle; vibration; camera lens contamination/obscuration; masking or interfering objects; strong light or shadow effects; low illumination; and further optical performance degradation. It is essential to address these factors and eliminate them during the data acquisition and processing stage.

The current work focuses on the usability of capsule networks compared with state-of-the-art convolutional neural networks. In this case, the authors' goal was to identify 15 keypoints of the face for further analysis. The authors compared the dynamic routing algorithm with the authors' proposed routing method. The networks were also trained using different adversarial attack techniques, highlighting the applied capsule-based solution's robustness. The research goal was that the authors' CapsNets solution should perform well in professional applications like DMSs in public transport. The first phase of the current study, presented here, was to improve the robustness of the face detecting algorithm, and further investigation is needed to monitor the speed and computational need of the system.

The current paper is structured as follows. In Section 2, the authors present the theory of the capsule network, discussing the authors' proposed optimization algorithm and activation function. The authors present the dataset used in their work and the network architectures developed. In Section 3, the authors clarify the conditions under which the networks were trained. The authors present the adversarial attack methods used in the training and the main parameters of the training process. The authors then present and explain the training results, comparing the different methods. In Section 4, the authors discuss the results and their importance, and finally, in Section 5, the authors summarize their work and give an outlook for future plans.

## 2. Methodology

The current study involves detecting specific parts of the human face. The primary aim is to find practical applications for the observation of drivers. In addition, the authors want to investigate the applicability and robustness of capsule networks in this area. This area is becoming more popular as technology develops, not only in research but also in an increasing number of practical applications, such as DMSs for bus drivers.

Verma et al. proposed a novel, real-time driver emotion monitoring system with face detection and facial expression analysis [42]. They used two VGG-16 neural networks. The first network extracts appearance features from the detected face image, and the second extracts geometrical features from the facial landmark points. Jain et al. proposed a capsule-network-based approach for detecting the distracted driver [43]. The proposed method performed well on real-world environment data, where the results were compared with convolutional-neural-network-based approaches.

Ali et al. generated a dataset for conducting different experiments on driver distraction [44]. They presented a novel approach that uses features based on facial points, especially those computed using motion vectors and interpolation to detect a special type of driver distraction. Liu et al. presented a comprehensive review of the research work on face recognition technology; they analyzed the difficulty of face recognition tasks, then introduced the main frameworks of face recognition from the geometric-feature-based, template-based, and model-based methods [40]. Their work compared different solutions and highlighted face recognition's importance and current limitations.

### 2.1. Capsule Network Theory

CapsNets are similar to neural networks but include some improvements on the known problems of neural networks [45,46]. A convolutional neural network cannot spatially map what is learned. For example, if it only ever sees images of an object from the same viewpoint, it will not recognize it from another viewpoint. Convolutional neural networks cannot discover the relationship between the elements of an image. For example, when detecting a human face, if the image contains features of a human face, it will detect it as a human face, even if the components of the face are not in the correct positions. Various

pooling layers are used in convolutional neural networks. These are generally efficient but tend to cause substantial loss of information by their nature. Capsule networks offer a more efficient solution to this by using a routing algorithm and vectors instead of scalar values, which allows them to store more information about a feature. The key difference between capsule and neural networks is the systems' basic building blocks. While neural networks are made up of neurons, capsule networks contain so-called capsules. Table 2 shows the main differences between classical artificial neurons and capsules.

**Table 2.** Differences between a capsule and a neuron.

| | **Capsule** | **Neuron** |
|---|---|---|
| Input | Vector ($u_i$) | Scalar ($x_i$) |
| Affine transform | $\hat{u}_{j|i} = W_{ij} u_i$ | - |
| Weighting | $s_j = \sum_i c_{ij} \hat{u}_{j|i}$ | $\alpha_j = \sum_i w_i x_i + b$ |
| Nonlinear activation | $v_j = \frac{\|s_j\|^2}{1+\|s_j\|^2} \frac{s_j}{\|s_j\|^2}$ | $h_j = f\left(a_j\right)$ |
| Output | Vector $\left(v_j\right)$ | Scalar $\left(h_j\right)$ |

A capsule can be considered a group of interconnected neurons that perform much internal computation and encapsulate the results of the computations into an $n$-dimensional vector. Where the vector is the output of the capsule, it can be seen that the output of the capsule is not a probability value, as is usual for a neuron. The length of the output vector gives the probability value associated with the output. The direction of the $n$-dimensional vector can store different features related to a given task. For example, a network is supposed to detect a human face. In this case, low-level capsules are responsible for facial components, such as the eyes, nose, or mouth, and a high-level capsule is responsible for face recognition. The neurons in the low-level capsules encode the object's internal properties, such as its position, orientation, color, texture, or shape.

In capsule networks, no pooling layer or similar solution is used for reduction between layers. In this approach, routing-by-agreement was used, where the output vector of any lower-level capsule is sent to all higher-level capsules. Each lower-level capsule output is compared with the actual output of the higher-level capsules. Routing aims to control the strength of the information flow between capsules. It is performed to achieve a stronger connection between related features.

Let $i$ be a lower-level capsule and $j$ a higher-level capsule. The $\hat{u}$ input tensor for capsule $j$ is calculated as:

$$\hat{u}_{(j|i)} = W_{ij} u_i \tag{1}$$

where $W_{ij}$ is a weighting matrix, initialized with random values, and $u_i$ is a pose vector for the $I$-th capsule. The coupling coefficients are calculated with a simple softmax function, as in the following:

$$c_{ij} = \frac{exp\left(b_{ij}\right)}{\sum_k exp\left(b_{ik}\right)} \tag{2}$$

where $b_{ij}$ is the log probability of capsule $I$ being coupled with capsule $J$ and $b_{ij}$ is initialized with zero values.

The total input to capsule $j$ is a weighted sum over the prediction vectors, as in the following:

$$s_j = \sum_i c_{ij} \hat{u}_{j|i} \tag{3}$$

Each layer of the capsule network gives output vectors. Therefore, the probability is determined by the length of the vector. A non-linear squashing activation function must be applied before determining the probability value. The squashing function is as follows:

$$v_j = squash(s_j) = \frac{\|s_j\|^2}{1 + \|s_j\|^2} \frac{s_j}{\|s_j\|^2} \qquad (4)$$

The routing algorithm is used to determine the relationship between the capsule layers. The dynamic routing algorithm (Algorithm 1) updates the $c_{ij}$ values and determines the output $v_j$ capsule vector.

---

**Algorithm 1.** Routing algorithm [46]

---

1:  **procedure** ROUTING $\left( \hat{u}_{j|i}, \, r, \, l \right)$
2:  for all capsule $i$ in layer $l$ and capsule $j$ in layer $(l + 1)$: $b_{ij} \leftarrow 0$
3:  **for** $r$ iterations **do**
4:  for all capsule $i$ in layer $l$: $c_i \leftarrow softmax(b_i)$
5:  for all capsule $j$ in layer $(l + 1)$: $s_j \leftarrow \sum_i c_{ij} \hat{u}_{j|i}$
6:  for all capsule $j$ in layer $(l + 1)$: $v_j \leftarrow squash\left(s_j\right)$
7:  for all capsule $i$ in layer $l$ and capsule $j$ in layer $(l + 1)$: $b_{ij} \leftarrow b_{ij} + \hat{u}_{j|i} v_j$
8:  **return** $v_j$

---

### 2.2. Proposed Capsule Routing Mechanism

Present experiments on capsule networks theory have shown that the $\hat{u}_{j|i}$ input in the dynamic routing algorithm has too large an impact on the output tensor. When calculating the output vector $v_j$, the formula includes the input $\hat{u}_{j|i}$ twice:

$$v_j = sq\left( \sum_i smax\left( b_{ij} + \hat{u}_{j|i} v_j \right) \hat{u}_{j|i} \right) \qquad (5)$$

where $sq(\cdot)$ is the squash function and $smax(\cdot)$ is the softmax function. To be able to improve the routing mechanism between lower- and upper-level capsules, the following modifications to the routing algorithm are proposed:

$$v_j = sq\left( \sum_i smax\left( b_{ij} + \sum_j \|v_j\| \right) \hat{u}_{j|i} \right) \qquad (6)$$

This minimal modification makes the routing algorithm simpler and faster to compute. The other proposed change concerns the squashing function. In the last capsule layer, a modified squashing function was applied, as follows:

$$squash(s) = \frac{s - e^{-\|s\|} s}{\|s\| + \varepsilon} \qquad (7)$$

where $\varepsilon$ is a fine-tuning parameter. The authors of the article have carried out several experiments with different hyperparameters. As a result, $\varepsilon = 1 \times 10^{-7}$ gave the best performance.

### 2.3. Dataset

The Kaggle Facial Keypoints Detection Dataset [47] was used in this work to test the novel solution, which aims to detect the location of keypoints on face images. The dataset contains a total of 7049 samples of different human faces. The dataset contains human faces in various situations, not specifically of bus drivers. However, this does not affect the results because the images in the dataset were taken with settings that could be implemented on board a bus. The samples have 15-15 output keypoints, which represent the position of the following parts of the face: left eye center, right eye center, left eye inner corner, left eye outer corner, right eye inner corner, right eye outer corner, left eyebrow inner end, left eyebrow outer end, right eyebrow inner end, right eyebrow outer end, nose

tip, mouth left corner, mouth right corner, mouth center top lip, and mouth center bottom lip. Unfortunately, the Kaggle dataset is incomplete; not all faces contain all 15 keypoints.

For this reason, only those samples were used that contain all the keypoints. Thus, a total of 2250 samples were used in the training process, with 1800 samples in the training set and 450 samples in the testset, where all images were 48 pixels in width and 48 pixels in height with 1 color channel. Figure 1 shows some samples from the dataset, where the keypoints are marked in red.

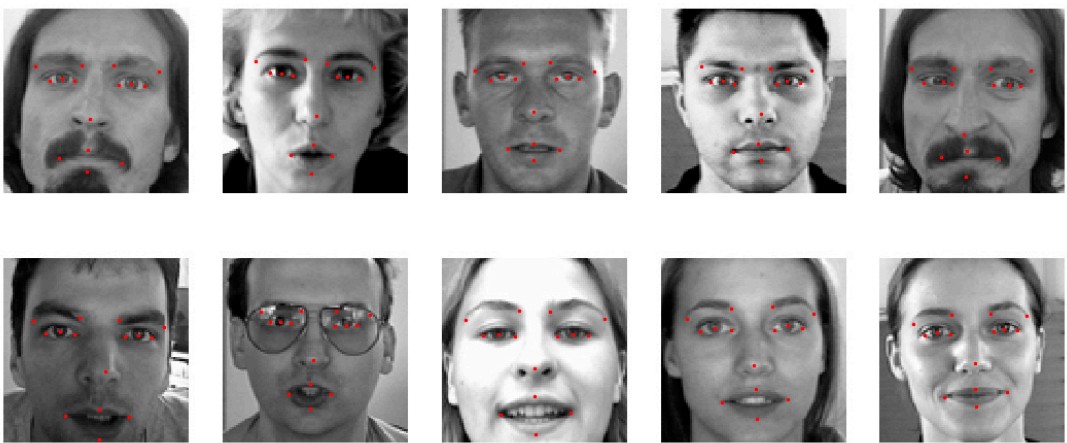

**Figure 1.** Samples from the Kaggle Facial Keypoints Detection Dataset [47].

*2.4. Network Architectures*

The primary goal of the current study was to focus on the differences between the proposed routing algorithm and the dynamic routing algorithm by Sabour et al. Second, the authors wanted to compare the effectiveness of capsule-based solutions with convolutional-neural-network-based approaches. In this research, four network architectures were used: the Capsule Network (CapsNet) proposed by Sabour et al. [46], a modified version of the Capsule Network with the proposed routing algorithm and activation function, the Wide-ResNet network proposed by Zagoruyko et al. [48], and the SimpleCNN network proposed by An et al. [49].

The authors' own capsule-based solution is based on the architecture of the original capsule network introduced by Sabour et al. [46]. The only differences are the routing algorithm and the activation function described in Section 2.1. The architecture of the authors' capsule network solution for this task is illustrated in Figure 2.

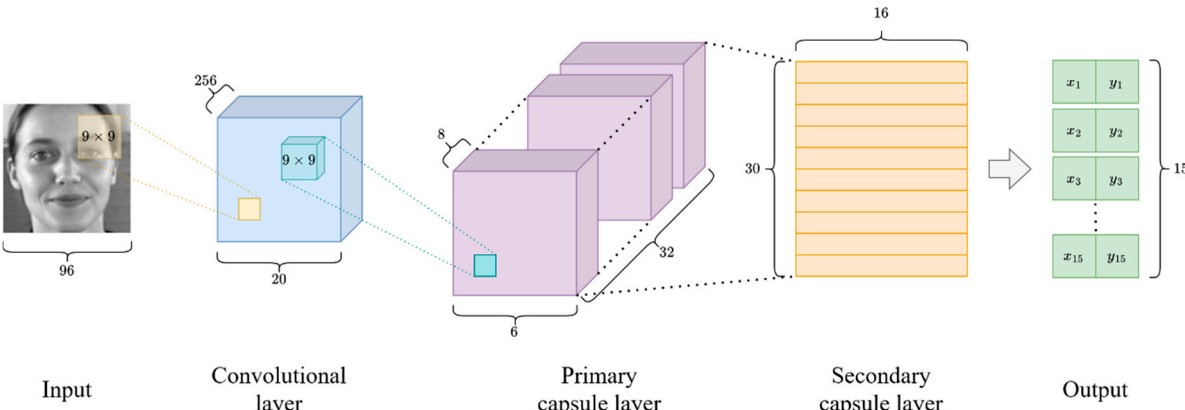

**Figure 2.** Capsule Network architecture.

The Wide-ResNet (Wide-Residual Network) architecture was introduced by Zagoruyko et al. [48]. The main building block of the Wide-ResNet network is the residual block:

$$x_{l+1} = x_l + \mathcal{F}(x_l, \mathcal{W}_l) \tag{8}$$

where $x_l$ is the input and $x_{l+1}$ is the output of the *l*-th layer, $F(\cdot)$ is the residual function, and $W_l$ is the parameter list of the residual block. The Wide-ResNet is based on a width parameter determined by factor *k* in the convolutional layers. In the original ResNet model, *k* = 1. However, in Wide-ResNet, the wideness of the network is determined by the *k* > 1 value. In this work, it has been shown that networks can be made more efficient by widening layers rather than creating deeper networks. In 2016, the Wide-ResNet architecture provided state-of-the-art results on several commonly used datasets, like Canadian Institute for Advanced Research-10 (CIFAR-10) [50], Canadian Institute for Advanced Research-100 (CIFAR-100) [50], Street View House Numbers (SVHN) [51], and Common Objects in Context (COCO) [52].

The Simple Convolutional Neural Network (SimpleCNN), introduced by An et al., consists of multiple convolution layers and a fully connected layer, where each convolution layer contains the following: 2D convolution, 2D batch normalization, and ReLU activation [49]. This neural network is made up of three sub-networks, where the sub-networks differ only in the convolutional layer kernel size. They used $3 \times 3$-, $5 \times 5$-, and $7 \times 7$-sized kernels. The final result of the three sub-networks is obtained by using majority voting. The SimpleCNN network is a state-of-the-art solution for image classification of the Modified National Institute of Standards and Technology (MNIST) dataset [53].

## 3. Training and Results

In the training process, adversarial attacking methods were applied to investigate the effectiveness of different neural and capsule networks. In this study, seven adversarial attacking methods were used on the presented two neural networks and two capsule networks.

### 3.1. Adversarial Attack Methods

The Fast Gradient Signed Method (FGSM) was proposed by Goodfellow et al. [54]. It was one of the first and most popular attacks to fool a neural network. The FGSM works by using the gradients of the neural network to create an adversarial example. The gradient value comes from the network's loss, and the algorithm modifies the input image with the loss gradient to maximize the loss value of the network as follows:

$$x_{adv} = x + \epsilon \times \text{sign}(\nabla_x J(\theta, \ x, \ y)) \tag{9}$$

where $x_{adv}$ is the adversarial image, $x$ is the original input, $y$ is the original label, $\epsilon$ is a parameter to ensure that the perturbations are small, $\theta$ is the model parameters, and $J(\cdot)$ is the loss function of the network.

Fast adversarial training using FGSM (FFGSM) was proposed by Wong and Rice et al. [55]. This method is very similar to the Fast Gradient Signed Method. The main difference is that this method uses random initialization, where random uniform distributed noise is added to the input image:

$$x' = x + \text{Uniform}(-\epsilon, \ \epsilon) \tag{10}$$

where $x$ is the input image, $x\prime$ is the noisy image, and $\epsilon$ is a parameter to ensure small perturbations.

Projected Gradient Descent (PGD) was proposed by Mądry et al. [56]. This approach is similar to the FGSM attack but iteratively executes the attack several times, as follows:

$$x'_{t+1} = \prod_{\mathcal{B}(x, \ \varepsilon)} \left\{ x'_t + \alpha \text{sign}\left( \nabla_{x'_t} J\left( f_\theta(x), f_\theta(x'_t) \right) \right) \right\} \tag{11}$$

where $\prod \mathcal{B}(x, \varepsilon)$ refers to the projection to $\mathcal{B}(x, \varepsilon)$ and $\mathcal{N}(0^n, I^n)$ is a normal distribution.

Two other variants of the PGD attack method were used. In the first case, noise generation was performed the same way as with the original PGD but with l2 loss (PGDL2). The next one was the TRADES' Projected Gradient Descent (TPGD) proposed by Zhang et al. [57]. In this method, the attacked examples generated by Projected Gradient Descent (PGD) with Kullback–Leibler divergence loss are as follows:

$$x'_0 = x + 0.001 \times \mathcal{N}(0^n, I^n) \tag{12}$$

$$x'_{t+1} = \prod_{\mathcal{B}(x, \varepsilon)} \left\{ x'_t + \alpha \text{sign}\left( \nabla_{x'_t} \updownarrow_{KL} \left( f_\theta(x), f_\theta(x'_t) \right) \right) \right\} \tag{13}$$

where $\prod \mathcal{B}(x, \varepsilon)$ refers to the projection to $\mathcal{B}(x, \varepsilon)$, $\mathcal{N}(0^n, I^n)$ is a normal distribution, and $\updownarrow_{KL}(\cdot)$ is a loss function [58].

The following attack method is based on a simple random noise generation. This approach adds Gaussian Noise (GN) to the input image. Here, the properties of the network, its current state, or the loss result are not used to generate the noise. The adversarial image is generated as follows:

$$x\prime = x + \sigma \mathcal{N}(0^n, I^n) \tag{14}$$

where $x$ is an input image, $\sigma \in (0, 1)$ is a fine-tuning parameter, and $\mathcal{N}(0^n, I^n)$ is a normal distribution value.

The Basic Iterative Method (BIM) was proposed by Kurakin et al. This work introduced a straightforward way to extend the Fast Gradient Signed Method (FGSM) [59]. The FGSM attack was applied multiple times with a small step size and clip pixel values of intermediate results after each step to ensure that they were in an $\epsilon$-neighbourhood of the original image. The attacked inputs were generated as follows:

$$x'_{x+1} = Clip_{x,\epsilon}\left\{ x'_n + \alpha sign\left( \nabla_x J\left( x'_n, y_{true} \right) \right) \right\} \tag{15}$$

where $x'_0 = x$, $x$ is the input image and

$$Clip_{x,\epsilon}\{x\prime\}(u, v, z) = min\{255, x(u, v, z) + \epsilon, max\{0, x(x, y, z) - \epsilon, x\prime(u, v, z)\}\} \tag{16}$$

is the clipping function, where $x(u, v, z)$ is the value of channel $z$ of the image $x$ at coordinate $(u, v)$.

### 3.2. Training Process

In this study, all four networks were trained in eight ways: in the first approach without using any adversarial attack technique, and then using the seven attack methods presented. In all cases, the training was performed under the same conditions with the same data. The training was carried out via the Paperspace cloud service, where all networks were trained under the same hardware conditions. An Nvidia RTX A4000 GPU was used in the training process. The Python programming language and the PyTorch deep learning framework were used for the implementation. The Adam (kingma2017) optimization algorithm was used for the training. The initial learning rate, based on empirical experience, was chosen to be 0.001. During the epochs, the learning rate was obtained as follows:

$$lr_{i+1} = lr_i \times 0.96^{\frac{1}{2000}} \tag{17}$$

where $lr_i$ is the learning rate in the $i$-th epoch and $lr_0 = 0.001$ is the initial learning rate. In each case, all networks were trained over 50 epochs. In the study, the efficiency for different epoch values was investigated; however, experience shows that the difference between the networks is already well visible here and does not change significantly in further iterations.

Batches of 16 sizes were used during the training and testing process. In this study, the following L1 loss was used. Let

$$\updownarrow(x, y) = \{l1, l2, \ldots, l_n\}^T \tag{18}$$

where $\ell$ is the loss of the given batch, $x$ is the predicted output, $y$ is the ground truth, and $n$ is the size of the batch. Let

$$l_i = \begin{cases} \frac{0.5(x_i - y_i)^2}{\beta}, & \text{if } |x_i - y_i| < \beta \\ |x_i - y_i| - 0.5\beta, & \text{otherwise} \end{cases} \tag{19}$$

### 3.3. Results

The efficiencies measured during training are shown in Figure 3. In the no-attack case, it can be seen that the two capsule networks show a similar curve, but their approach shows a visibly smoother, more uniform learning. In the end, the authors' own solution led to better results. The Wide-ResNet network performs slightly worse than the authors' capsule-based solution. The SimpleCNN network converges more slowly initially but eventually outperforms all networks.

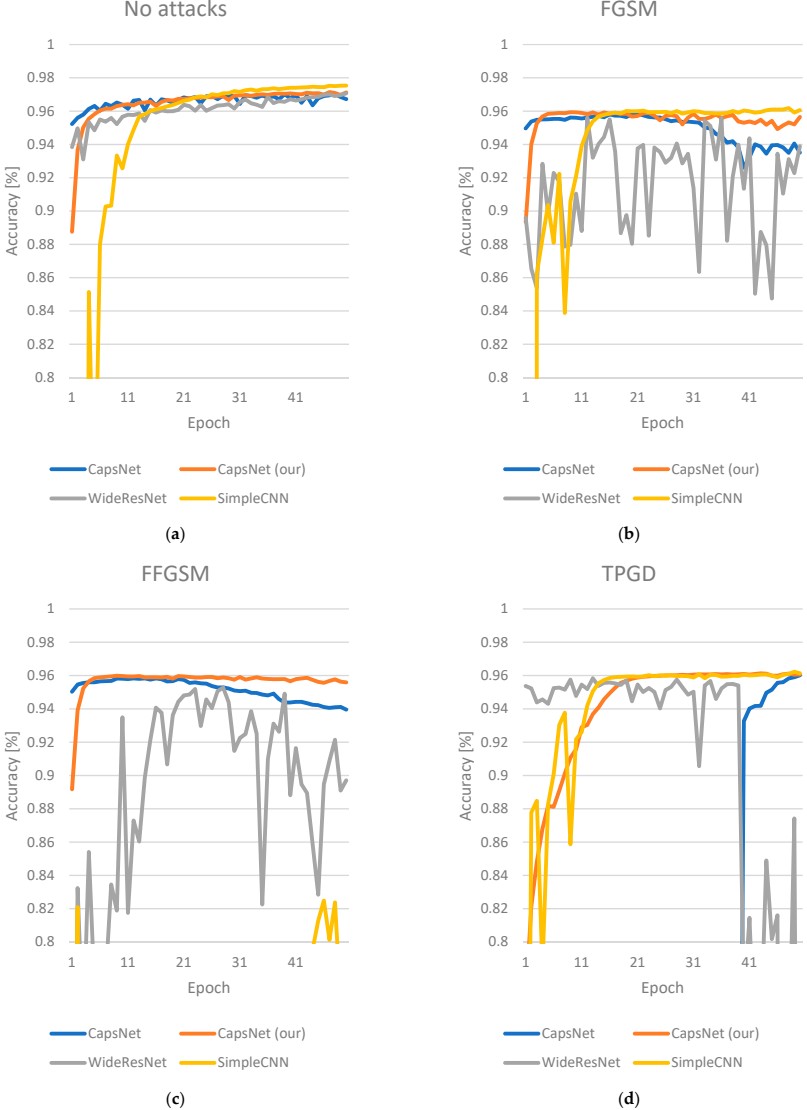

**Figure 3.** *Cont.*

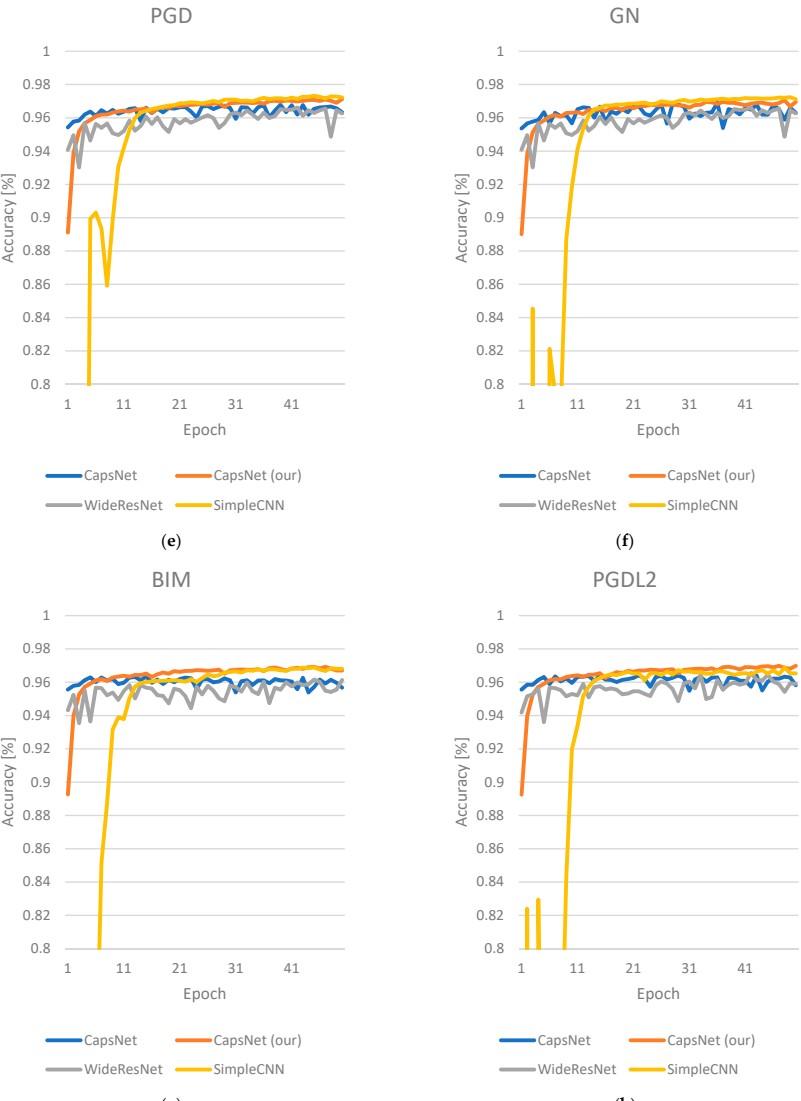

**Figure 3.** The effectiveness of networks in the training process under different attack methods (**a**—no attack; **b**—FGSM; **c**—FFGSM; **d**—TPGD; **e**—PGD; **f**—GN; **g**—BIM; **h**—PGDL2).

The results are a little different for the attack methods. The most considerable degradation was observed for the Wide-ResNet network. In several cases, the network failed to converge to an optimal state at all. The SimpleCNN network typically performed well, tending to be more affected by noise only at the beginning of the learning phase. Capsule-based solutions generally performed well. In the solution of Sabour et al., the training process started well in some cases, but by the end, the efficiency of the network had decreased. The current proposed capsule-based approach also typically performed well. A similar phenomenon can be observed as with the original capsule solution. For some attack methods, minimal degradation was observed by the end of the learning process. However, in all cases, the authors' capsule-based solution with their routing mechanism performed better than the original with the dynamic routing algorithm.

Table 3 summarizes the best results achieved during the training. It can be seen that the SimpleCNN network, as a state-of-the-art solution, performed outstandingly in most cases. However, the currently proposed method performed best for three of the seven attack methods tested. Only for the capsule networks, the authors' method always performs better.

**Table 3.** The effectiveness of each network under different attack methods (the bold numbers mean the highest values in each column).

|  | No Attack | FGSM | FFGSM | TPGD | PGD | GN | BIM | PGDL2 |
|---|---|---|---|---|---|---|---|---|
| Wide-ResNet | 0.9707 | 0.9561 | 0.9528 | 0.9584 | 0.9661 | 0.9661 | 0.9616 | 0.9655 |
| SimpleCNN | **0.9753** | **0.9618** | 0.8248 | **0.9623** | **0.9732** | **0.9724** | 0.9689 | 0.9689 |
| CapsNet | 0.9703 | 0.9582 | 0.9583 | 0.9602 | 0.9678 | 0.9691 | 0.9635 | 0.9643 |
| CapsNet (own, current) | 0.9714 | 0.9593 | **0.9599** | 0.9613 | 0.9712 | 0.9706 | **0.9693** | **0.9698** |

If someone looks at how the effectiveness of each network has changed compared to itself under different attack methods, different results can be obtained. Table 4 shows by what percentage the efficiency decreased for each attack method compared to the case without attack. It can be seen that the capsule networks always performed best in this case. The authors' solution performed better in most attack methods tested.

**Table 4.** The rate of degradation of network efficiency for different attack methods (the bold numbers mean the lowest percentages in each column).

|  | FGSM | FFGSM | TPGD | PGD | GN | BIM | PGDL2 |
|---|---|---|---|---|---|---|---|
| Wide-ResNet | 1.50% | 1.84% | 1.27% | 0.47% | 0.47% | 0.93% | 0.53% |
| SimpleCNN | 1.39% | 15.43% | 1.34% | 0.22% | 0.30% | 0.66% | 0.66% |
| CapsNet | **1.24%** | 1.24% | **1.04%** | 0.25% | 0.12% | 0.70% | 0.61% |
| CapsNet (own, current) | 1.25% | **1.19%** | **1.04%** | **0.03%** | **0.09%** | **0.23%** | **0.16%** |

## 4. Discussion

The authors have presented their proposed optimization algorithm for capsule networks in this study. To support the validity of the authors' own algorithm, they conducted tests where the authors' solution was compared with capsule networks trained with the dynamic routing algorithm, as well as with several state-of-the-art convolutional neural network solutions [54–59]. In the comparison, different adversarial attack techniques were used to test the robustness of the presented systems. The results show that capsule-based solutions can compete with state-of-the-art convolutional neural-network-based solutions in this task. It can be said that when adversarial attack methods are used, capsule networks perform well. The authors' results show that the degradation in efficiency is noticeably smaller for capsule networks than for the presented convolutional neural networks. For the case of capsule networks, the authors' proposed method achieved better performance than the dynamic routing algorithm in all cases. Regarding performance degradation, the authors' solution performed better in six out of seven cases. In terms of effectiveness, the proposed method was more effective in three out of seven cases.

From an application perspective, given the growing need for safety in public transport, a critical point for a sufficiently secure image-based driver monitoring system is the elimination of distracting or offensive incidents, the attacks. The reliability of data recording depends mainly on external factors, such as vibration, camera lens contamination, lighting conditions, and other optical performance degradations. The authors' CapsNets method reduces the uncertainty of data recording to increase data processing accuracy.

This new approach has further potential for improvement. Besides effectiveness, speed and computational demand should also be examined to minimize hardware needs. All on-board DMSs require high-speed, real-time data processing, low power consumption, flexible connectivity, and robustness, for both software and hardware. The current research will further deepen the investigations into the algorithm's applicability while striving to design a modular system, thus aiding the feasibility of implementation.

However, the current solution is not yet perfect, and its practical application is not recommended in its current state. The authors' long-term goal is to have an efficient and reliable solution as part of the custom driver monitoring system, which is currently being

implemented. To be able to achieve this, it is necessary to improve further the current capsule-based solution to achieve even higher efficiency. It will require further testing on an even wider range of datasets and testing the robustness of the network with additional attack methods. Finally, measuring the effectiveness and validating the system on real data from the end-user environment is necessary.

## 5. Conclusions

As wearable monitoring devices are unsuitable for monitoring drivers in public transport, reliable software solutions must be developed for face recognition.

The current study focuses on developing a capsule-network-based approach for face detection to support DMS applications for safer road traffic. This paper presents the proposed new routing algorithm for capsule network optimization. We have shown that the introduced method is suitable for detecting keypoints of the human face. Different adversarial attack techniques were used to test the robustness of the systems. The study results show that the efficiency of capsule networks is, in many cases, higher than the neural networks presented, and the authors' proposed routing method achieved better performance than the dynamic routing algorithm in all cases. Regarding performance degradation, the authors' solution performed better than most.

The algorithm's outstanding performance warrants further investigation to develop an application-level implementation for a driver monitoring system for use in public road transportation. The authors' computational vision method's proven robustness can eliminate the disruption of external factors, such as vibration, camera lens contamination, lighting conditions, and other optical performance degradations.

This paper presents the preliminary findings of an extensive, ongoing research and development endeavor. In the following research phase, we want to build on these results. We want to validate the effectiveness of the networks on additional datasets. We would like to generate our own dataset of bus drivers, where drivers can be observed in real traffic situations. This will allow us to investigate specific edge cases and make the network capable of detecting them. We would like to optimize further our current solution pending further studies. We would like to compare the results obtained with other approaches on a broader scale, also in the application of further adversarial attack techniques.

**Author Contributions:** Conceptualization, J.H. and V.N.; methodology, J.H. and V.N.; software, J.H.; validation, J.H. and V.N.; formal analysis, J.H. and V.N.; investigation, J.H. and V.N.; resources, J.H. and V.N.; data curation, J.H. and V.N.; writing—original draft preparation, J.H., Á.B., G.K., S.F. and V.N.; writing—review and editing, J.H., Á.B., G.K., S.F. and V.N.; visualization, J.H.; supervision, J.H., Á.B., G.K., S.F. and V.N.; project administration, J.H., S.F., and V.N.; funding acquisition, J.H., S.F. and V.N. All authors have read and agreed to the published version of the manuscript.

**Funding:** This research received no external funding.

**Data Availability Statement:** All data used in this research are presented in the article.

**Acknowledgments:** The authors wish to acknowledge the technical support received from the Vehicle Industry Research Centre and Széchenyi István University, Győr. The research was carried out as part of the Cooperative Doctoral Program supported by the National Research, Development and Innovation Office and the Ministry of Culture and Innovation.

**Conflicts of Interest:** The authors declare no conflict of interest.

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
