# Peer review of "Face Detection Using a Capsule Network for Driver Monitoring Application"

_computers, doi:10.3390/computers12080161_

Round 1

Reviewer 1 Report

Dear authors,

I am glad I had the opportunity to review your manuscript. Overall, I found it interesting and well written, and I have only minor suggestions, that I think, will be easily tackled by you.

I have only one more important comment, that is a concern for me. In the paper you presented this new capsule network, that seems to work better on face recognition than other methods, and you therefore suggest to use it for bus drivers’ face recognition. Nevertheless, I do not see a concrete relationship with bus drivers in your research. Was the dataset used in your research composed of bus drivers’ faces? Or did you test the network on other data related to bus drivers? Otherwise, I don’t think you should mention this issue, because your method can be probably be applied to many other fields. In this case, also the introduction should be changed and fitted to a wider context.

Abstract:

Line 9: “Driver distraction sources and complex physical and psychological effects must be recognized and analyzed in real- world driving conditions” – why? I suggest to add a brief explanation, e.g. “in order to reduce the risk related to these factors, …”

Line 10: The camera-based system – I suppose there are more of these systems – please generalize.

And why is it the best monitoring method? How can you assess it? I suggest you to keep the whole sentence more general, e.g.: “one of the most applied/best working/… monitoring methods is…”

Lines 19-20: I suggest to add here that you applied the network to 7 case studies and that in 6 out of 7 it outperformed the other networks, which it was compared with.

Line 20-21: it is tricky to finish the abstract with this sentence, since it leaves the reader with some expectations for a further clarification. I suggest it to reformulate it.

Key-words: I would suggest to add “bus driver”, since the method is applied to this group of traffic users.

Introduction:

Line 26: “The need for safe public transport is becoming more demanding for drivers as traffic density increases”. From a traffic point of view, it is not only related to traffic density: it is related also to traffic mix, to the number of different and fast external stimula, especially present in urban area, where the majority of busses are travelling…

Line 27: “Advanced driver-assistance systems (ADAS) can partly intervene in emergencies, e.g., braking [1], but the driver's task and responsibility remains to control the vehicle safely.”. True. Additionally, it should be consider that ADAS have also some side effects, e.g. drivers being  distracted by checking them. Nowadays there is a number of information within the car that each driver should deal with and that cognitively loads him/her.

And this is specifically true for bus drivers. I suggest you to improve this part by shifting lines 30-32 after the brief accident data analysis for bus drivers. In this way, the whole introduction will be much more connected and clearer.

Line 48: after “were identified as the main causes” there is the need for a reference.

Line 63: “Most research on vigilance focuses on sleep deprivation”. Please, cite the sources.

Lines 65-66 and 66-67 need references.

Line 122: “This approach is more convenient for the drivers” – why?

Lines 130-140 need references.

Materials and Methods – maybe the title can be adapted to “methods” or “methodology”

Sub-section 2.1 is very much appreciated.

Maybe I would suggest to add the general advantages/disadvantages of capsule networks in comparison to neural networks.

Line 220: “The authors' experiments on capsule networks theory have shown that…” is it meant in general, or the experiments you worked out for this research? Maye it would be better to first explain how you defined your capsule network (so, the dataset, the reasons for the choice of a formulation rather than another, etc.) and then to explain your proposed mechanism.

Also, in lines 229-230 previous authors’ research could be cited, to stand behind this sentence.

Line 232: is this sample related to bus drivers? Or is it a general sample?

Figure 3. I suggest to use different type of lines, to allow a better distinction also when the paper is printed black & white.

Line 363: “However, the currently proposed method performed best for three of the seven attack methods tested” – it seems to me, that this sentence is in contrast, with what you wrote in the abstract (that is probably related to the efficiency point of view). Please, elaborate better on this point.

I do not have comments on that.

Author Response

See the attached PDF file in which we put a comparison part at the end to be able to show all the changes tracked.

Reviewer 2 Report

The research paper focuses on detecting parts of the human face with the purpose of observing and detecting emotion and activity in bus drivers. The overall aim is to use tis system  to improve the safety and lower the risk of accidents by real-time monitoring of bus driver face, using a camera and capsule networks.

The paper presents relevant state-of-the-art considering related domains that are involved in dealing with the issue. The capsule network algorithms is used with a specific routing algorithm, while adversarial attacking methods are applied in the training and validation phase. The effectiveness and degradation rate of each network was measured under no or several types of network attacks, proving the proposed method is the most robust and effective in most cases.

In terms of improvement I would recommend a comparison of the results with reference to the literature related presented in section 1 (so not just methos, but also numerical results).

Also, in terms of training and evaluation please provide more details on this phase, for example, training performance (system specification, time complexity etc.). Also, please justify the choice of number of epochs in each case, why it was selected as such and if there would be expected a different behaviour in other contexts which is recommended for future work or less likely to be relevant.

The language is academic, domain specific and appropriate for the purpose of the paper. The references are various, covering a wide timeframe, perhaps reconsider those before 2015 if they are all relevant to the paper. Please consider some missing information/wrong formatting in references 3, 8, 20, 41, 42 etc. and check the journal formatting requirements.

Overall, the paper presents novel results with applications in a relevant and interesting domain. I recommend the paper for publication, but I strongly recommend revising the above mentioned points.

Author Response

(The authors gave the same response as above.)

Reviewer 3 Report

The material of the paper is new and innovative.

The authors have carried out a good job in explaining their ideas and organizing their material.

Accordingly

The abstract is fully informative about the content of the paper.

The introduction, although quite extensive, constitutes a very good introduction and simultaneously an adequate review of the topic.

The second section is very well substantiated with the required mathematics.

Section three should be subdivided into the proper subsections.

The conclusions should be enlarged and more descriptive.

The references are very well supporting the paper and they are UpToDate.

Author Response

(The authors gave the same response as above.)

Round 2

Reviewer 1 Report

Dear authors,

I am glad to see that you agreed with my previous comments and that you improved those points. 

I do not have further suggestions.

Best regards

-